# Small-Scale Anisotropy in Stably Stratified Turbulence; Inferences Based on Katabatic Flows

**Eliezer Kit** [1,*] and **Harindra J. S. Fernando** [2,3]

1    School of Mechanical Engineering, Tel Aviv University, Tel Aviv 69978, Israel
2    Department of Civil and Environmental Engineering and Earth Sciences, University of Notre Dame, Notre Dame, IN 46530, USA; harindra.j.fernando.10@nd.edu
3    Department of Aerospace and Mechanical Engineering, University of Notre Dame, Notre Dame, IN 46530, USA
*    Correspondence: elikit@gmail.com or kit@tauex.tau.ac.il; Tel.: +972-3-6408-929

**Abstract:** The focus of the current study is on the anisotropy of stably stratified turbulence that is not only limited to large scales and an inertial subrange but also penetrates to small-scale turbulence in the viscous/dissipation subrange on the order of the Kolmogorov scale. The anisotropy of buoyancy forces is well-known, including ensuing effects such as horizontal layering and pancakes structures. Laboratory experiments in the nineties by Van Atta and his students showed that the anisotropy penetrates to very small scales, but their experiments were performed only at a relatively low $Re_\lambda$ (i.e., at Taylor Reynolds numbers) and, therefore, did not provide convincing evidence of anisotropy penetration into viscous sublayers. Nocturnal katabatic flows having configurations of stratified parallel shear flows and developing on mountain slopes provide high Reynolds number data for testing the notion of anisotropy at viscous scales, but obtaining appropriate time series of the data representing stratified shear flows devoid of unwarranted atmospheric factors is a challenge. This study employed the "in situ" calibration of multiple hot-film-sensors collocated with a sonic anemometer that enabled obtaining a 90 min continuous time series of a "clean" katabatic flow. A detailed analysis of the structure functions was conducted in the inertial and viscous subranges at an $Re_\lambda$ around 1250. The results of DNS simulations by Kimura and Herring were employed for the interpretation of data.

**Keywords:** stratified shear flow; structure function; direct numerical simulation (DNS); intermittency; skewness

## 1. Introduction

Stably stratified turbulence, as already discussed in many previous publications (e.g., [1–3]), is an intriguing phenomenon. The more capabilities there are to delve into smaller scales of stratified turbulence, the more it unravels the inherent complexity [4]. Stably stratified turbulence is ubiquitous in the ocean and atmosphere, and, on larger scales, both stratification and the Earth's rotation become important, leading to geostrophic turbulence. These external effects were discussed by Lumley [5] in their seminal paper, which also considered the effects of an external magnetic field and shear in causing departure from Kolmogorov's regime. It is quite obvious that external anisotropic forces lead to the anisotropy of turbulence, a conspicuous example being the magnetic lines associated with an external magnetic field that lead to directional effects. Columnar vortices with axes parallel to magnetic lines do not naturally interact with magnetic lines, thus curbing the Joule dissipation and increasing the survivability of such vortices. Such behavior was observed in MHD experiments [6], opening up the possibilities for creating two-dimensional turbulence in the laboratory. Attempts were made to employ an analogous approach to strongly (stably) stratified flows. It is unclear, however, why columnar vertical vortices should be more supplant. These vortexes are subject to zig-zag instabilities [7,8], leading

to their collapse and the development of thin horizontal layers containing pancake vortexes. These layers, together with the thin layers separating the horizontal layers, form cells of height $L_v = u'/N$, where $L_v$, $u'$, and $N$ are the characteristic vertical scale, RMS longitudinal turbulence velocity, and the Brunt–Väisälä frequency, respectively. Instead of longitudinal velocity $u'$, the RMS of vertical velocity $w'$ is often used. It is possible that the scaling in this case is strongly dependent on the above characteristic vertical scale.

Similar behavior could be expected when shear acts as an external forcing, although the flow situation is essentially different, since the thin layers, above and below the horizontal layers, are more prone to instability. The emergence of layers in a pure shear flow is, apparently, less plausible. Layering with a long horizontal extent is also not expected in flows with external rotation.

When laboratory grid-generated homogenous isotropic turbulence (HIT) [1] is subjected to (thermally generated) stable stratification, turbulence becomes *anisotropic*, and the relevant controlling parameter of the overall flow is the inverse of the internal/turbulent Froude numbers, $Fr = \varepsilon/Nu'^2$, where $\varepsilon$ is the rate of dissipation. The above expression can be modified by assuming that $\varepsilon = u'^3/L_h$ and using $Fr = u'/(NL_h)$, where $L_h$ is the horizontal length scale. The typical inverse Froude numbers assessed in our field experiments, dubbed MATERHORN, were $2 \div 5$ [9], coincidentally in the same range as that observed in [10], though the $Re_\lambda$ is larger, thus allowing for some comparisons. Additional references are given in the discussion chapter.

## 2. Methodology

In the nocturnal Atmospheric Boundary Layer (ABL), stable stratification evolves overnight due to radiative cooling, thus leading to interesting small-scale phenomena that was studied during MATERHORN (2011–2016) field experiments [11]. A novel probing system was deployed to capture very small scales, down to Kolmogorov viscous dissipation subrange [12,13]. The finer scales therein were captured by multiple hot-film probes placed in the probe volume of a sonic anemometer, with the latter providing low frequency data to calibrate the former by utilizing a neural network. This sonic- and hot-film anemometer dyad (dubbed the "combo" probe) was placed on a horizontal pole at 6 m height of a 32 m high tower (labeled ES-2), equipped with an array of sonics and thermocouples at various levels; see [12]. ES-2 had seven levels of sonics: 0.4, 4, 10, 16, 20, 25, and 28 m. Careful analysis of data enabled identification of periods that fit the rubric of stratified parallel shear flows, which emerged as katabatic (downslope) flows draining from the nearby Granite Mountain.

After careful processing of sonic records of ES-2, which was a part of a densely instrumented flux tower array (ES-1–ES-5) designed to study katabatic flows at night on the east slope of Granite Mountain (for details see [11,12]), a 90 min period starting from 22:00 MDT (Mountain Daylight Time (local time)) on October 19, 2012 was selected for stratified shear flow studies. Prior to the selected time interval, the wind speed rapidly increased from ~1 m/s to ~4 m/s at the combo probe height. The wind direction changed from its usual oscillations before 22:00 MDT to a nearly constant direction, resembling a stratified parallel flow with relatively low shear. Thereafter, the winds changed quite rapidly between 23:30 MDT and midnight [12]. Although mean quantities were quasi-steady during the selected period, 22:00–23:30 MDT, careful inspection of turbulence statistics showed considerable variability. An analysis of approximately homogeneous subintervals, identified based on stability (*Fr*), enabled focusing on structure function analysis in the subinterval $SI_{devd}$ (22:50–23:10 MDT). This subinterval was considered as fully developed turbulence, since flow variations were modest, and appearance of small-scale bursting events [12] was very limited (less than 2% of the time).

### 3. Stratified Turbulence in Field Experiments and DNS

#### 3.1. Puzzling Observations in the MATERHORN Campaign

The detailed results concerning nocturnal turbulence observed during the MATER-HORN campaign were published in a series of papers [5,11,12]. In these studies, different data analyses and presentation methods were employed, viz., the spectral approach in Fourier space [12] and the structure functions approach in physical space [4]. Although both approaches are inter-related, each of them better reflects specific features of nocturnal turbulence. In particular, a small-scale bursting phenomenon was uncovered using the spectral approach [12], and the celebrated Kolmogorov $-5/3$ spectrum appropriately described the measured spectral behavior in the inertial subrange for so-called no-bursting intervals, where "bursts" were effectively removed from the time series using proper thresholds based on KE dissipation. At the same time, the structure functions approach enabled the discovery of inertial and viscous (dissipation) subintervals with separate, corresponding power exponents for the bursting and no-bursting intervals. In this paper, we limit ourselves to the time series where the bursting events are removed.

In particular, our recent paper [4], on the stably stratified turbulence occurring in the nocturnal turbulence observed during MATERHORN [11,12], reported an unexpected behavior of the canonical (normalized) third-order longitudinal structure function (Figure 1).

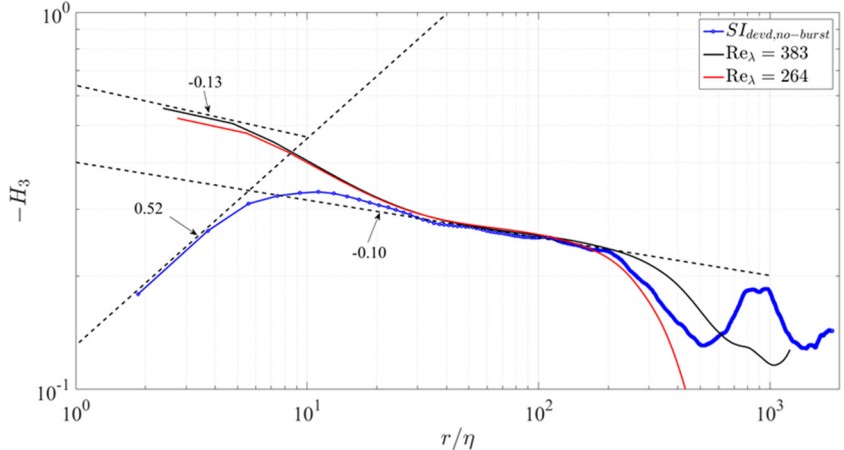

**Figure 1.** Comparison of the canonical third-order moments obtained in the field campaign [11,12] at $Re_\lambda$ = 1250 and DNS for the second-highest $Re_\lambda$ = 264 and 383.

In Figure 1, the canonical third-order structure function measured at a 6 m height from the ground was compared with the same structure functions evaluated for nearly isotropic turbulence obtained by classical box DNS computations [14]. In the field, the $Re_\lambda$ was 1250, whereas in the DNS computations the $Re_\lambda$ was 264 and 383. While in the inertial subrange ($r/\eta = 20 \div 200$), both the qualitative and quantitative agreements were very good, in the viscous subrange ($r/\eta \leq 10$) the results were conspicuously different, and the difference increased with the decrease in separation. It is worth noting that at small normalized separations $r/\eta < 10$, the scaling exponents were even of different signs for the field and DNS data. Since, in the viscous subrange, one can expect linear dependence between velocity and separation, e.g., [15], the scaling exponent for the conventional third-order structure function should be about zero to satisfy Kolmogorov's Self-Similar Hypothesis (KSSH) [16].

This difference of scaling exponents in the viscous subrange of field experiments and DNS is puzzling, especially when there is a clear agreement among the scaling exponents in the inertial subrange with each other and with the Kolmogorov inertial subrange scaling. To shed light on this perplexing behavior, the canonical normalized third-order structure function was computed at various separations; it is defined as the third-order structure function $L_3$ divided by the second-order structure function $L_2$ to the power 1.5, e.g., $H_3(r) = L_3(r)/(L_2(r))^{3/2}$ , which also represents the skewness of the structure func-

tion. In Figure 2, we separately consider the third- and second-order structure functions, both normalized according to KSSH [16].

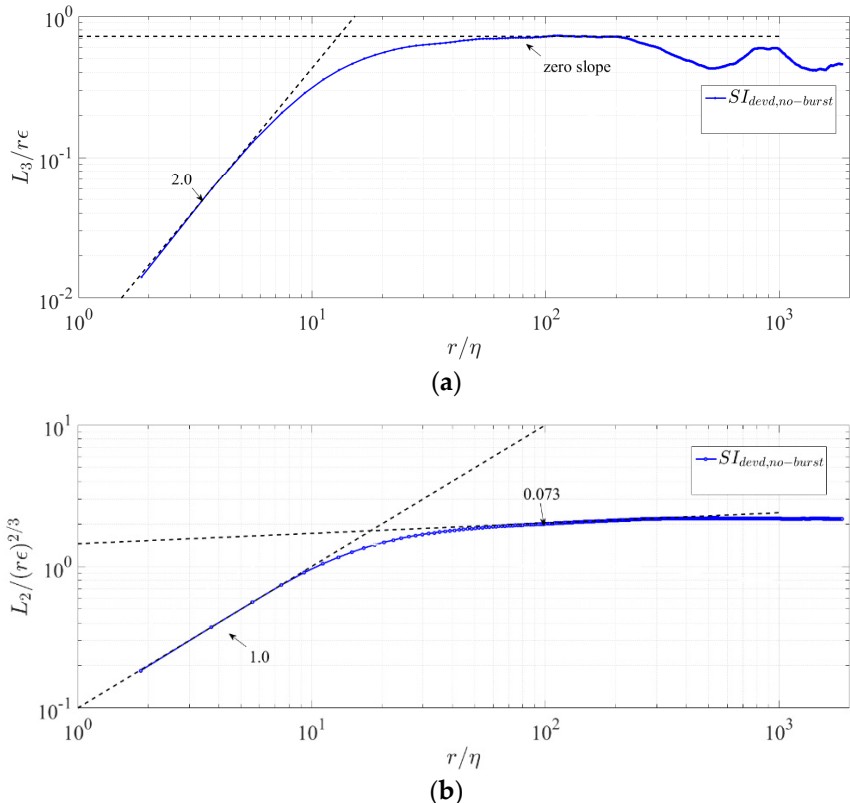

**Figure 2.** Normalized according to KSSH third-order (**a**) and second-order (**b**) structure functions obtained from the field data, computed using longitudinal velocity increment $\Delta u(x,r)$.

The third-order structure function in Figure 2a yields, in the viscous subrange, the expected linear dependence of the characteristic velocity increment $\Delta u(x,r)$ on the separation $r$, e.g., $\Delta u(x,r) \sim r^1$. However, the second-order structure function in Figure 2b corresponds to a different type of dependence, namely, $\Delta u(x,r) \sim r^{5/6}$. This explains the scaling exponent observed for the canonical third-order structure function in Figure 1.

However, as shown in Figure 1, the power exponent of the canonical structure function is relatively small, $\sim -0.1$ in the DNS computations of the approximately homogeneous isotropic turbulence, compared to $\sim +0.5$ obtained in the field experiment. The determination of the separate scaling exponents from the DNS data, in a manner similar to that of the field data, as shown in Figure 2, yielded the following dependencies, $\Delta u(x,r) \sim r^{0.87}$ for third-order and $\Delta u(x,r) \sim r^{0.91}$, for second-order structure functions. It is worth noting that the expectation of identical behavior by odd and even structure functions is generally unjustified, despite it being widely quoted and often used, in particular, in the Extended Self-Similarity (ESS) approach. In previous research [4], we found that all even (second, fourth, and sixth in this study) structure functions yield $\Delta u(x,r) \sim r^{5/6}$ in the viscous subrange. Then, it was found that all the odd structure functions (first, third, and fifth in this study) constructed using the modulus of velocity increment $|\Delta u(x,r)|$ yield a similar relation, namely, $\Delta u(x,r) \sim r^{5/6}$ in the viscous subrange; here, $\Delta u(x,r)$ is the characteristic velocity increment at selected separation $r$. The consistent and significant difference between the power exponents of the third- and second-order structure functions is responsible for the substantial power exponent 0.5 at the small scales of the conventional third-order structure function/skewness. In isotropic DNS, this power exponent is about $-0.1$, which is much closer to zero, though of a different sign. This result is somewhat perplexing, given the expectation that, due to the local isotropy at small scales, the power exponent tends to be zero. Even more puzzling is that, in the inertial subrange, the behavior of the canonical

third-order structure function in the DNS and field experiments shows full (qualitative and quantitative) agreement.

This apparently contradicts with the postulate of local isotropy (PLI), which entails better isotropy whence the separation ($r/\eta$) is diminishing. Cambon et al. [17], however, predicted such behavior in cases where external anisotropic forces such as rotation (Coriolis) and stable stratification (buoyancy) do not produce turbulent energy. In particular, they state the following: "It is often considered that anisotropy only affects the largest scales of the turbulent flow, so that the cascade of anisotropy only mildly penetrates toward the scales in the inertial range, with eventually no direct impact on the dissipative range, according to the view inherited from Kolmogorov [16]. In this scheme, the large-scale anisotropy induced by body forces and/or mean gradients is just considered as a particular modality of 'forcing' the largest scales. This viewpoint is radically questioned for flows in which the body force responsible for anisotropy is energy conserving." Moreover, they state that "Anisotropy develops from isotropic initial conditions due to nonlinear energy transfer towards the plane of zero frequency in wave-vector space. However, two-dimensionalization does not occur: the fraction of energy at or near the zero-frequency plane remains small at all times. The cascade to small scales is strongly anisotropic, producing angle-dependent spectra which become more and more (not less) anisotropic, the smaller the scale considered".

Obviously, in shear flows, turbulent energy is produced, notwithstanding, the laboratory experiments with homogeneous shear flow by Warhaft and their co-workers [18–20] unambiguously showed that the return to isotropy expected at the small scales does not occur either at a lower $Re_\lambda \sim O(100)$ [18] or at a higher $Re_\lambda \sim O(1000)$ [19], which was again ascribed to shear penetrating to smaller scales. To quote [19], "The results show that PLI is untenable, both at the dissipation and inertial scales, at least to $R_\lambda \sim 1000$, and suggest it is unlikely to be so even at higher Reynolds numbers." See more considerations in the Discussion section.

Our results [4] appear to strongly support the observation that stratified turbulence differs from nearly isotropic turbulence, even at small scales corresponding to the dissipation (viscous) sublayer, and raise a question about the PLI for this important case. It was very desirable to assert the validity of our results by independent measurements at a higher $Re_\lambda \sim O(1000)$ or the DNS computations of stably stratified turbulence.

Unfortunately, the laboratory experiments of stably stratified turbulence are limited to a relatively low $Re_\lambda$ (<100), and, thus, high-quality field experiments with continuous measurements at high sampling rates are called for. In fact, during the entire MATERHORN campaign, only a few records provided "clean" data sets for the nocturnal stably stratified turbulence strongly affected by thermal stratification. It should be stressed that these measurements were possible due to our novel calibration approach that enabled in situ calibration of a multi-hot-film probe based on the simultaneous measurements of low-frequency 3D-velocity data of a collocated sonic anemometer. Employing machine learning (neural network training) enabled the calibration of the hot-film probe in situ, thus avoiding problems with the hot-film's potential deterioration in hostile field environments. The efficacy of this calibration method [21] was tested in a series of papers [22–24] and proved to be very efficient, even in the presence of noise to some degree.

In the current work, the neural network procedure was based on the Multi-Layer Perceptron (MLP) approach. MLP contains one input layer, one or two hidden layers, and one output layer. The number of nodes of the input layer and the number of input signals are the same. We used a fully connected network. In this case, each node of the input layer duplicates and sends its input signal to every neuron of the first hidden layer. The hidden layer consists of a number $h$ of neurons. For the training of MLP, the Back Propagation and Conjugate Gradient Descent methods were sequentially used. In all cases, the number of epochs that was necessary for the generation of the neural network did not exceed 100.

The use of two collocated x-probes (oriented in the horizontal and vertical planes) at a very small separation allowed for the measurement of the full 3D velocity vector. Via the Taylor hypothesis, the time series of the velocity components could be converted into a

velocity series in the mean wind direction. As expected, the processed data records were limited in time (about 1 min) to keep the mean transverse velocity components as small as possible. Directly after the end of the record, the probe was aligned with the mean horizontal velocity that was evaluated during the last 5 s of the record. Each record was flagged as successful if the mean wind direction differed by less than 10° from its value at the beginning of the record.

Stratified turbulence is essentially anisotropic (at least at large scales) and is characterized by horizontal pancake-like layering; thus, we posit that the surprising result observed in Figure 1 is related to anisotropy and layering. The DNS of the stably stratified turbulent flow in a box could help in this regard, notwithstanding its idealized nature. DNS simulations were conducted by Jack Herring in collaboration with Yoshi Kimura and were presented in two seminal papers, one in the *Journal of Fluid Mechanics* [2] and the other in *Physica Scripta* [25]. The significance of their work is limited not only to the very detailed and comprehensive presentations of the spectra and structure functions obtained at a relatively high $Re_\lambda$ but also to the elegant methodology of velocity data analysis based on Craya–Herring decomposition. The latter approach allows for the separation of the entire oscillating flow into 2 types of modes: horizontal and *vortical*; vertical and *wavy* [2,23]. In Section 3.2, the puzzling results (the penetration of the anisotropy caused by stratification into small scales) of our study are discussed in light of [2].

### 3.2. DNS of Stably Stratified Turbulence by Kimura and Herring 2012 [2]

As shown in [2], the second-order structure function $St_2(d, u, x)$ for the longitudinal velocity component $u$ at separation $d$ ($r$ in our notations) in the $x$-direction can be presented as the superposition of two terms following Craya–Herring decomposition (expression 4.4 in [2]).

$$St_2(d;u,x) = 2\pi \int\limits_{-\infty}^{\infty} dk_z \int\limits_{0}^{\infty} k_\perp dk_\perp [\Phi_1(k_\perp,k_z)\left(1 - 2\frac{J_1(k_\perp d)}{k_\perp d}\right) +$$
$$+ \frac{3k_z^2}{k_\perp^2+k_z^2}\Phi_2(k_\perp,k_z)\left(1 - \frac{4}{3}J_0(k_\perp d) + \frac{2}{3}\frac{J_1(k_\perp d)}{k_\perp d}\right)]$$

where $\Phi_1$ and $\Phi_2$ are defined as energy densities; the $k_\perp$ and $k_z$ components of vector number $\boldsymbol{k}$ are in the horizontal plane and the vertical direction, respectively; and $J_0$ and $J_1$ denote the Bessel function of the corresponding order. While $\Phi_1$ contributes only to velocities in the horizontal plane and is determined by vertical vorticity component $\omega_z$, $\Phi_2$ is contributing to both horizontal and vertical velocity components and can be determined using only the vertical velocity component $w$, as presented in expressions 2.9 and 2.10, respectively [2]. Since, in a stratified flow, the vertical direction coincides with gravitational acceleration $g$, the second term of the decomposition may account for the internal waves. Indeed, the behavior of the second-order structure function due to the first term only, evaluated in [2], is practically the same as that of the isotropic turbulence (see Figure 14 in [2]), while adding the second term accounting for buoyancy leads to behavior of the second-order structure function that substantially differs from that of the isotropic turbulence.

The second-order longitudinal structure function in the $x$-direction (see Figure 15a in [2]) is shown below in Figure 3. Unfortunately, the authors do not use normalization that follows KSSH for data presentation as it is made in our Figure 2b. Notwithstanding, it is clear that, in the inertial subrange, the scaling exponent is greater than Kolmogorov's value of 2/3, which is similar to our result. With the addition of buoyancy effects (the second term), the viscous subrange scaling exponent for the velocity increment's dependence on the separation is close to 5/6 (see the estimated slope 5/3 for the second-order structure at low separations). Unfortunately, [2] does not present the third-order structure function, which could adumbrate Kolmogorov's 4/5 law in the inertial subrange for the horizontal longitudinal third-order structure function of stratified turbulence.

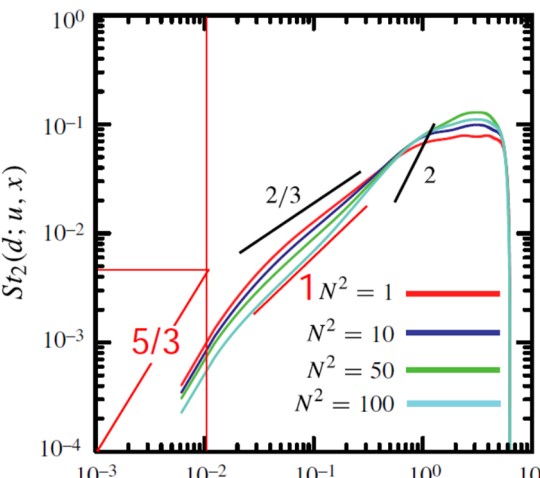

**Figure 3.** Second-order structure functions for squared normalized buoyancy frequency $N^2$ = 1, 10, 50, and 100 as a function of separation *d*. The assessed power exponent in viscous subrange is about 5/3. (Reproduced with permission from Kimura and Herring [2], *J. Fluid Mech.* 698, 19. Copyright 2012, Cambridge University Press.)

In recent studies [26,27], the DNS of rotating turbulence in a box was reported as enabling the derivation of the third-order structure functions at different polar angles for examination of their shapes, in particular the penetration of anisotropy to scales in the viscous subrange. It is important to emphasize that rotation, similarly to stratification, leads to modifications, from isotropic to axisymmetric, of the turbulence structure, by generating columnar vortices around the rotation axis. The resemblance is, obviously, not exact and does not include horizontal layering by pancake structures in the horizontal plane, as predicted and observed in stably stratified turbulence. A mechanism of the formation of pancake structures from columnar vortices in stratified turbulence, as mentioned above, is often related to zig-zag instability.

An interesting result observed in the Ph.D. research of Vallefuoco [27] is the development of the clear anisotropy of third-order moments as a function of the polar angle. While DNS without rotation yielded perfect isotropy with moments for polar angles *0* and $\pi/2$ that practically coincided in both inertial and viscous subranges, when rotation is present the moments differ across the scales, including at very small scales on the order of the Kolmogorov scale (see Figure 4.31a,c in [27]).

### 3.3. A Simplified Model

Inspired by the derivations and results of [2], as presented in Section 3.2., we suggest an (over)simplified model. This model assumes that the longitudinal velocity $u(x)$ measured at a 6 m height consists of two contributions: $u_1(x)$, representing pure HIT (isotropic and asymmetrical), and $u_2(x)$, the strongly stratified turbulence (anisotropic and symmetric, with respect to the vertical), with a low correlation between the two. The PDF of $u_1$ is essentially asymmetrical for the longitudinal velocity derivative, while the PDF of $u_2$ may be assumed to be symmetrical. It follows that the third-order structure function for $\Delta u(x, r)$ is determined by $u_1$ only.

From below expression for $L_3(r)$, following above assumptions,

$$L_3(r) = \langle (\Delta u(x, r))^3 \rangle = \langle (\Delta u_1(x, r))^3 + 3(\Delta u_1(x, r))^2 \Delta u_2(x, r) + 3\Delta u_1(x, r)(\Delta u_2(x, r))^2 + (\Delta u_2(x, r))^3 \rangle$$

it can be easily assessed that, at the right hand side, the second and third terms are zero, due to the lack of correlation between $u_1$ and $u_2$, and the fourth term is zero due to the symmetry of the probability density function of $u_2$, leaving the first term as the nonzero term. The situation is different for the higher-order odd longitudinal structure functions.

For example, in the fifth-order structure function, the first $\langle(\Delta u_1(x,r))^5\rangle$ and the third terms $10 * \langle(\Delta u_1(x,r))^3\rangle * \langle(\Delta u_2(x,r))^2\rangle$ are nonzero. This simplified model, therefore, may explain the intriguingly different behaviors in the *viscous* subrange of the odd third- and fifth-order structure functions for $\Delta u(x,r)$. That is, the expected linear dependence of velocity increment on separation $r$ for the third-order structure function (Figure 5a in [4]) and an oscillating slope (scaling exponent) for the fifth-order structure functions (Figure 8a in [4]).

The odd first-, third-, and fifth-order structure functions evaluated for $|\Delta u(x,r)|$ in the *viscous* subrange yield the same $r^{5/6}$ dependence (i.e., scaling exponent $p*5/6$; Figure 9 in [4]) as *all* the even structure functions. It is obvious that all the structure functions for the absolute velocity increments include both contributions ($u_1$ and $u_2$). Therefore, the scaling exponent, in general, can be different from that of the homogeneous turbulence. Relatively weak anomalies only start to appear at $p = 6$. However, presently, we are unable to offer a sound explanation for the distinct shape (5/6 scaling exponent) in the above dependence.

Kimura and Herring [2] obtained a surprising result: the ratio of potential energy to kinetic energy for *all N* is about the same, about 0.1. According to [2], "This potential energy is attributable to the three-dimensional turbulence that occupies the space between the quasi-horizontal layers." They note the following: "Why this value asymptotes to ~0.1 of the kinetic energy is a mystery to us." Our belief is that the characteristics of these quasi-horizontal layers are of the utmost importance for the strongly stratified turbulence in the viscous subrange, so the appropriate answer to each one of the above questions might be the key to the mystery. The low Reynolds number laboratory experiment by Fincham et al. [28] illustrates that the interactions between the turbulence in the layers and strong shear between (highly dissipative) layers are characteristics of decaying stratified turbulence, similar to the observations noted by Métais and Herring [29].

The model offered above is simplistic, so a more rigorous attempt to separate isotropic and anisotropic contributions can be made in the framework of *SO*(3) formalism [30], by conducting the decomposition of the appropriately measured structure or correlation functions into spherical harmonics. This is left for future work.

## 4. Discussion

Our study concerns the anisotropy of stably stratified turbulence that is limited not only to large scales and inertial subrange but also to penetrating the (smaller-scales) domain of the viscous/dissipation subrange on the order of the Kolmogorov scale. The anisotropy of buoyancy forces as well as their propensity for developing horizontal layering and pancakes structures are widely known and may be related to this anisotropy across the scales.

In a recent work [31], direct numerical simulations were performed for the assessment of the local isotropy of dissipative scales for stably stratified flows. It is claimed that the estimation of the dissipation rate of turbulent kinetic energy and density variance can be approximated using the isotropic assumption for the turbulent Froude number $Fr \geq \mathrm{O}(1)$. There is a clear departure from isotropy for $Fr < \mathrm{O}(1)$. However, in another work [32], it was shown that dissipation-scale isotropy is determined by $\mathrm{Re}_b$, the buoyancy Reynolds number, only. The authors also performed direct numerical simulations to investigate the anisotropy of stratified turbulence and the transition to isotropy at small length scales. Turbulence was generated by forcing large-scale vortical modes, an approach that is broadly consistent with geophysical stratified turbulence. The authors' results suggest that $\mathrm{Re}_b \geq 500$ is required to obtained the same degree of small-scale isotropy seen in the unstratified turbulence at a similar Re.

In relatively old but very comprehensive study [33] in 1981, the authors have already performed direct numerical simulations of decaying homogeneous turbulence in density-stratified fluids. They have examined in particular the energetics, the evolution of characteristic length scales and the importance of nonlinearities in the computed flow fields. They have found that stratification introduces wave-like characteristics into the flow fields. This is exemplified by exchange from (and to) kinetic energy to (and from) potential

energy and the development of counter-gradient buoyancy fluxes. Stratification tends to enhance the growth of horizontal scales while inhibiting the growth of vertical scale.

Per a referee, we would also like to point out the very interesting work [34] by Smith and Waleffe, with the title "*Generation of slow large scales in forced rotating stratified turbulence*", but their focus is on large scales in forced rotating stratified turbulence, so it appears to have a lesser relevance to our work. Indeed, the authors discussed the cases of pure stratified turbulence that was ***randomly forced at small scales***. Their interests are mainly related to the inverse cascade. The same remarks are applicable to the comprehensive study of Delache et al. [35], which dealt with anisotropy in freely decaying rotating turbulence.

Following the comments of reviewers, we carefully evaluated various options for the modeling generation of anisotropic small-scale structures. To do so, we referred to additional studies [36–38] such as Kerstein 1999, Wunsch 2000, and Wunsch and Kerstein 2001 that describe the horizontal layering in stably stratified turbulence, which suggested a one-dimensional turbulence (ODT) model for this analysis. In essence, this model locally applies mixing length theory throughout the simulation domain, defining a wide range of possible mixing lengths and corresponding time scales for each point in space. Turbulent mixing is then randomly applied throughout the system at all length scales, based on the locally appropriate time scales. It is a stochastic model and can be very useful in applications.

Finally, we refer to the most up-to-date monograph [39], which summarized the recent theoretical, computational, and experimental results dealing with homogeneous turbulence dynamics. A large class of flows was covered: flows governed by anisotropic production mechanisms (e.g., shear flows) and flows without production but dominated by waves (e.g., homogeneous rotating or stratified turbulence).

While laboratory experiments in the 1990s showed the penetration of anisotropy to very small scales, such experiments were performed at a relatively low $Re_\lambda$ and, therefore, did not provide convincing evidence of anisotropy penetration into the viscous subrange. Measurements in the atmosphere during the MATERHORN project could provide such evidence as discussed in this paper, where the viscous subrange was accessed via a specialized (combo) probe, which is an assembly of a high (space-time)-resolution, multi-sensor hot-film probe array collocated with a sonic that measures the full velocity vector at a low-frequency resolution.

By employing machine learning based on a neural network algorithm during post-processing, "in situ" calibration of the combo probe was accomplished. "Clean" stratified turbulence data unblemished by other effects were observed in the Katabatic flows developing near the mountain slopes during nocturnal events, which can provide appropriate data for stratified turbulent studies. As explained in Section 2, 90 min of clean stratified shear flow data could be gleaned, based on which detailed structure function analyses were conducted in the inertial and viscous subranges. Anisotropic behavior in the viscous subrange was revealed at an $Re_\lambda$ around 1250, which can be considered substantial. The seminal paper of Kimura and Herring [2] on the DNS of stably stratified turbulence was employed for data interpretation.

Our results strongly support that stratified turbulence differs from nearly isotropic turbulence even at a smaller dissipation (viscous) subrange. To the best of our knowledge, this is the first time that such inferences are made at a relatively high Taylor-based Reynolds number of $Re_\lambda$ ~1250. This result contradicts the hypothesis of the postulate of local isotropy (PLI), which posits enhanced isotropy, whence the separation ($r/\eta$) is diminishing. In fact, Cambon et al. [17] predicted kindred behavior when external anisotropic forces, such as rotation (Coriolis) and stable stratification (buoyancy), do not produce turbulent energy.

The Craya–Herring decomposition used in [2] allowed for presenting the second-order structure function for the longitudinal velocity increment at separation $r$ in the $x$-direction as a superposition of two terms: the turbulence determined by vertical vorticity $\omega_z$ and the internal waves determined by vertical velocity component $u_z$. While the former is

essentially isotropic on the horizontal plane, the latter term is responsible for anisotropy, which we used to qualitatively interpret the observations.

**Author Contributions:** Both authors equally contributed to the paper. All authors have read and agreed to the published version of the manuscript.

**Funding:** This work was funded in part by grant number N00014-21-1-2296 (Fatima Multidisciplinary University Research Initiative) of the Office of Naval Research (ONR), administered by the Marine Meteorology and Space Program. The MATERHORN program was funded by ONR award number N00014-11-1-0709. The complex terrain flows research of the authors is also funded by the US National Science Foundation, award number AGS-1921554.

**Institutional Review Board Statement:** Not applicable.

**Informed Consent Statement:** Not applicable.

**Conflicts of Interest:** The authors declare no conflict of interest.

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
