# Peer review of "Small-Scale Anisotropy in Stably Stratified Turbulence; Inferences Based on Katabatic Flows"

_atmosphere, doi:10.3390/atmos14060918_

Round 1

Reviewer 1 Report

See my report under pdf format to be uploaded

Reviewer 2 Report

The sentence is easy to understand and the logic is clear

Author Response

We were asked not to respond to Reviewer 2, just to other two Reviewers.

We provided above point-by-point responses to the Reviewers 1 and 3.

Reviewer 3 Report

Using experimental measurements of atmospheric turbulence collected during the “MATERHORN” project, the authors present evidence suggesting that stably-stratified turbulence exhibits anisotropy down within the smallest-scales of the flow (the viscous subrange). Their experiments build upon previous suggestions of such an effect but which had been limited by a focus on relatively low Taylor-Reynolds numbers as compared to the larger Taylor-Reynolds number of approximately 1250 considered here. Notably, the results provide contractidory evidence to the traditional hypothesis that turbulent motions are isotropic at the smallest-scales. 

The work is interesting and I would recommend publication in Atmosphere provided the authors consider my comments below.

  1. A diagram illustrating details such as the methodology, location at which the turbulence was measured, background flow etc. as part of the MATERHORN experiment would be useful in Section 2. 
  1. A brief summary of details about the machine learning algorithm used to post-process the experimental data would be useful (~line 200).
  1. A variety of other studies have also considered small-scale anisotropy in stably-stratified turbulence. It would be prudent to provide references of such studies in a more detailed introduction, and to compare their findings with the present analysis. For example, see: Garanaik and Venayagamoorthy, Physics of Fluids, 2018 (https://doi.org/10.1063/1.5055871); Lang and Waite, Physical Review Fluids, 2019 (https://doi.org/10.1103/PhysRevFluids.4.044801) among others.

  2. Similarly, have other experimental campaigns of atmospheric turbulence found similar effects to the ones you report here?

A careful proofreading of the document is required. In the abstract alone, there are a number of grammatical errors: 

“The focus of THE current study”,

“order of THE Kolmogorov scale”;

“pancake structureS”;

“convincing evidence (no S)”;

“90-minute SPACE continuous”. 

Other typos (not a complete list):

MATRHORN mis-spelled in Section 3.1 title.
